# Dendropanoxide, a Triterpenoid from *Dendropanax morbifera*, Ameliorates Hepatic Fibrosis by Inhibiting Activation of Hepatic Stellate Cells through Autophagy Inhibition

**DOI:** 10.3390/nu14010098

**Published:** 2021-12-27

**Authors:** Yong-Joo Park, Dong-Min Kim, Hye-Been Choi, Mi-Ho Jeong, Seung-Hwan Kwon, Ha-Ryong Kim, Jong-Hwan Kwak, Kyu-Hyuck Chung

**Affiliations:** 1College of Pharmacy, Kyungsung University, Busan 48434, Korea; yjpark@ks.ac.kr; 2School of Pharmacy, Sungkyunkwan University, Suwon 16419, Korea; kdm9947@gmail.com (D.-M.K.); ly0412b@skku.edu (H.-B.C.); 3New Drug Research Center, Handok Inc., Seoul 06235, Korea; 4Center for Systems Biology, Massachusetts General Hospital, Boston, MA 02114, USA; algh8906@naver.com; 5Department of Neurology, Johns Hopkins University School of Medicine, Baltimore, MD 21205, USA; kwon4862@naver.com; 6College of Pharmacy, Daegu Catholic University, Gyeongsan 38430, Korea; kimhr@cu.ac.kr

**Keywords:** dendropanoxide, hepatic fibrosis, hepatic stellate cells, autophagy, carbon tetrachloride

## Abstract

Hepatic fibrosis results from chronic liver damage and is characterized by excessive accumulation of extracellular matrix (ECM). In this study, we showed that dendropanoxide (DPX), isolated from *Dendropanax morbifera*, had anti-fibrotic effects on hepatic fibrosis by inhibiting hepatic stellate cell (HSC) activation. DPX suppressed mRNA and protein expression of α-SMA, fibronectin, and collagen in activated HSCs. Moreover, DPX (40 mg/kg) treatment significantly lowered levels of liver injury markers (aspartate aminotransferase and alanine transaminase), expression of fibrotic markers, and deposition of ECM in a carbon tetrachloride-induced mouse model. Anti-fibrotic effects of DPX were comparable to those of silymarin in a hepatic fibrosis mouse model. As a possible mechanism of anti-fibrotic effects, we showed that DPX inhibited autophagosome formation (LC3B-II) and degradation of p62, which have important roles in HSC activation. These findings suggest that DPX inhibits HSC activation by inhibiting autophagy and can be utilized in hepatic fibrosis therapy.

## 1. Introduction

Hepatic fibrosis is a chronic liver disease characterized by increased production and deposition of extracellular matrix (ECM) and is a global health burden [1]. Hepatic fibrogenesis is a dynamic process that can lead to disruption of the normal cellular function of the liver [2]. During hepatic fibrogenesis, hepatic stellate cells (HSCs), portal myofibroblasts, and immune cells are important in ECM production in response to liver injury. HSC activation is a key event in hepatic fibrogenesis and is responsible for about 80% of total fibrillar collagens in the fibrotic liver [3,4].

HSCs account for 5–10% of all hepatic cells, are present in the perisinusoidal space of Disse, and store vitamin A under normal healthy conditions [5]. Hepatic injury transforms HSCs to myofibroblast-like cells that show proliferative, contractile, and migratory properties and secrete pro-inflammatory and pro-fibrotic mediators [5]. Alpha-smooth muscle actin (α-SMA) is a specific marker for activated HSCs, and increased α-SMA-positive cells are observed in hepatic fibrosis patients [6,7,8,9]. Therefore, inhibition of HSC activation has been considered an important strategy in the development of drugs for hepatic fibrosis [5,10]. Autophagy is a catabolism process that has cell-specific roles during chronic liver diseases [11]. In HSC, autophagy promotes fibrogenic properties and collagen accumulation; therefore, autophagy inhibition in HSCs can be a therapeutic target for hepatic fibrosis [12]. 

*Dendropanax morbifera* Léveille (DPM), a subtropical evergreen tree belonging to the family Araliaceae, is distributed and cultivated in the southern regions of South Korea [13]. DPM has been widely used as a traditional medicine, and the Korea Food and Drug Administration (KFDA) has approved the leaf, stem, and root of DPM as a food ingredient [13,14]. DPM has anti-oxidant [14,15], anti-inflammatory [16,17], anti-diabetic [18,19], and anti-atherogenic properties [20]. The aqueous extract of DPM was recently shown to prevent thioacetamide-induced hepatoxicity and oxidative stress in rats, but the anti-fibrotic effects of DPM components have not been investigated in detail [21].

In this study, we aimed to discover an anti-fibrotic agent for hepatic fibrosis by inhibiting HSC activation. Dendropanoxide (DPX) was isolated from the ethanolic extract of DPM, and its anti-fibrotic effect was analyzed in activated LX-2 human hepatic stellate cells and a carbon-tetrachloride-induced hepatic fibrosis mouse model. The molecular mechanisms of DPX were explored, and we evaluated the therapeutic potential of DPX based on anti-fibrotic effects.

## 2. Materials and Methods

### 2.1. Extraction and Isolation of DPX

Aerial parts of *D. morbifera* were collected in February 2019 from a farm in Goheung-gun, Jellanam-do, Korea (34°48′05″ N and 127°18′06″ E). The plant was identified by author Dr. J.H. Kwak, a specialist in plant taxonomy. A voucher specimen (specimen No.: SKKU-Ph-19-002) was deposited at the Herbarium of School of Pharmacy, Sungkyunkwan University. The dried aerial parts of DPM (2 kg) were extracted twice in 95% ethanol (8 L) at 18–22 °C for 24 h and then once at 60 °C for 5 h. The total filtrate was concentrated under reduced pressure at 40 °C to obtain an ethanolic extract. This extract (73.57 g) was suspended in distilled water (0.9 L) and successively partitioned with dichloromethane (CH_2_Cl_2_), ethyl acetate (EtOAc), and n-butanol (n-BuOH) to give CH_2_Cl_2_ (19.03 g), EtOAc (2.91 g), n-BuOH (6.50 g), and water (57.05 g) fractions. A portion (8.3 g) of the CH_2_Cl_2_ fraction was subjected to silica gel column chromatography (5 × 42 cm) using a stepwise elution with hexane-CH_2_Cl_2_ (10:1–5:1–3:1–1:1, *v*/*v*; 500 mL for each solvent mixture), CH_2_Cl_2_ only (500 mL), CH_2_Cl_2_-MeOH (50:1–10:1–1:1, v/v; 500 mL for each solvent mixture), and MeOH only (1 L) to yield 9 fractions (F1–F9). The F3 fraction (hexane-CH_2_Cl_2_ = 3:1 fraction, 952 mg) was twice chromatographed over silica gel columns (3 cm × 40 cm) using hexane-CH_2_Cl_2_ solvent mixtures (3:1 and 1:1, respectively) to obtain a white crude compound (806 mg). The crude compound was further purified by recrystallization using the CH_2_Cl_2_-MeOH solvent mixture (1:3, *v*/*v*) to afford DPX (735 mg; Rf: 0.22, SiO_2_, hexane-EtOAc = 5:1, *v*/*v*, 10% H_2_SO_4_ in EtOH) at 97.6% purity by GC–MS analysis.

### 2.2. Chemicals

Transforming growth factor (TGF)-β1 was obtained from R&D Systems (#240-B; Minneapolis, MN, USA), and chloroquine (#C6628) and silymarin (#S0292) were obtained from Sigma-Aldrich (St. Louis, MO, USA).

### 2.3. Cell Lines and Culture

LX-2 cells, immortalized human hepatic stellate cells, were obtained from the ATCC (Manassas, VA, USA), and 10% fetal bovine serum (Biotechnics Research Inc., Lake Forest, CA, USA) with Dulbecco’s Modified Eagle’s Medium (Sigma) was used to support cell growth. Cells were maintained at 37 °C in an atmosphere (95% air and 5% CO_2_) under saturation humidity.

### 2.4. Animal Experiments

Animal experiments were approved by the Sungkyunkwan University Animal Care Committee (SKKUIACUC2018-10-44-2). Six-week-old male C57BL/6 mice (22–25 g) were purchased from Daehan BioLink (Chungbuk, Korea). After acclimatization for one week, the mice were randomly divided into six groups (seven mice per group) that received: (1) olive oil (vehicle control) and PBS; (2) carbon tetrachloride and PBS; (3), (4), (5) carbon tetrachloride and DPX (2, 10, 40 mg/kg, respectively); and (6) carbon tetrachloride and silymarin (40 mg/kg). All chemicals were treated with intraperitoneal injection (I.P.). Mice were treated with carbon tetrachloride (2 mL/kg) three times per week until experiment completion. After three weeks of carbon tetrachloride injection, mice were treated with PBS or DPX (2, 10, 40 mg/kg) or silymarin (40 mg/kg) three times per week for three weeks. On day 43, mice were anesthetized with isoflurane and liver tissue, and blood samples were collected and stored at −80 °C before analysis.

### 2.5. Liver Histology and Blood Analysis

The formalin-fixed liver tissues were embedded in paraffin and cut into sections (4 µm). Hematoxylin and eosin (H&E) and Masson’s trichrome stains were used for analysis. Twenty randomly selected images of each stained section were used for quantification. Alanine transaminase (ALT) and aspartate aminotransferase (AST) levels in mouse serum were analyzed by Beckman Coulter AU680 (Beckman Coulter, Brea, CA, USA) as a measure of liver function (ChemOn Inc. Suwon, Korea).

### 2.6. Cell Viability Assay

The WST-1 assay was used to assess cell viability (Roche, Mannheim, Germany). After placing LX-2 cells in 96-well plates (1 × 10^4^ cells/well) for 24 h, cells were treated with DPM extract or DPX for 48 h. After adding WST-1 reagent, cells were incubated for 30 min at 37 °C. The absorbance at 440 nm and 690 nm was measured using a microplate spectrophotometer for analysis (Molecular Devices, Sunnyvale, CA, USA). 

### 2.7. Comparative Quantitative Real-Time PCR (qRT-PCR)

TRIzol reagent was used to isolate total RNA (Life Technologies, Grand Island, NY, USA). After cDNA was synthesized by the High-Capacity cDNA Reverse Transcription System (Life Technologies), qRT-PCR was performed using SYBR^®^ Premix Ex TaqTM (Life Technologies) and CFX96 Real-Time PCR System (Bio-Rad, Hercules, CA, USA). qRT-PCR was conducted using the primers listed in Table 1 (Bioneer, Daejeon, Korea). GAPDH was used to normalized mRNA expression levels.

### 2.8. Western Blot Analysis

After placing LX-2 cells in 6-well plates (5 × 10^4^ cells/well) for 24 h, chemicals were treated for 48 h after TGF-β1 treatment (48 h). After washing with PBS, cells were lysed with radioimmunoprecipitation assay buffer (Thermo Scientific, Waltham, MA, USA) with a protease inhibitor cocktail (GenDEPOT, Barker, TX, USA). Protein samples were denatured and resolved by 10% or 8–16% gradient SDS-PAGE (Bio-Rad) and transferred to a polyvinylidene difluoride membrane (Bio-Rad). After blocking with skim milk, the membrane was incubated overnight with primary antibodies. After washing with TBS-T, membranes were incubated with secondary antibodies conjugated to horseradish peroxidase. To detect the protein bands, an enhanced chemiluminescence reagent (Bio-Rad) was used. The GAPDH band was used to normalize the relative expression of the protein of interest. Anti-fibronectin (ab2413, Abcam, MA, USA), anti-collagen 1 (ab138492), anti-alpha-SMA (ab5694, Abcam), anti-p-SMAD2 (#3108, CST, Danvers, MA), anti-p-p38 (#9211), anti-p-ERK (#4370), anti-p-AKT (#9271), anti-β-catenin (#8480), anti-p62 (#2775), anti-LC3B (#8025), and anti-GAPDH (015-25473, Wako Pure Chemical Industries, Osaka, Japan) were used in Western blot analysis.

### 2.9. Statistical Analysis

The data were analyzed using GraphPad Prism version 7.00 (GraphPad Software Inc., San Diego, CA, USA). All experiments were performed at least three times. The data are expressed as mean ± standard deviation (SD). The differences between the groups were evaluated by Duncan’s post hoc test after one-way analysis of variance (ANOVA). 

## 3. Results

### 3.1. Effects of DPM Extract on Activated Hepatic Stellate Cells

LX-2 cells were treated with nontoxic concentrations of DPM extract (0–80 μg/mL) for 48 h after TGF-β1 pre-treatment (2 ng/mL, 48 h) (Figure 1A). TGF-β1 was used to activate LX-2 cells, and expression of α-SMA and collagen was measured by Western blotting. TGF-β1 treatment increased α-SMA and collagen protein expression, and DPM extract significantly inhibited their expression (Figure 1B).

### 3.2. Isolation of DPX and Its Effects on Activated Hepatic Stellate Cells

DPX (Figure 2) was obtained as a white solid from the dichloromethane fraction of DPM ethanolic extract. DPX was isolated and purified by repeated silica gel column chromatography with various solvent mixtures and recrystallization using the CH_2_Cl_2_-MeOH solvent mixture (Appendix A). This compound exhibited spectroscopic data of ^1^H NMR, ^13^C NMR, and GC–MS (Appendix A) consistent with the literature values [22].

To determine the appropriate treatment concentration of DPX, cytotoxicity of various concentrations of DPX (12.5–100 μg/mL) was measured after incubation of the cells with DPX for 48 h (Figure 3A). To identify the inhibitory effects of DPX on HSC activation, 0.1–2 μg/mL of DPX application for 48 h did not show cytotoxicity after activation by TGF-β1. As shown in Figure 3B, protein expression of α-SMA, fibronectin, and collagen was significantly increased after activation by TGF-β1; however, DPX blocked this increase. Similarly, mRNA expression of *α-SMA*, *collagen 1A1*, *collagen 3A1*, and *fibronectin* was alleviated by DPX in a concentration-dependent manner (Figure 3C).

### 3.3. Anti-Fibrotic Effects of DPX in Hepatic Fibrosis Mice 

We performed the experiment with a carbon tetrachloride-induced hepatic fibrosis mouse model to test the effects of DPX on hepatic fibrosis. As shown in Figure 4A, DPX (2, 10, and 40 mg/kg) was administered intraperitoneally for three weeks at three weeks of carbon tetrachloride I.P. injection. Silymarin (40 mg/kg) was administered as a positive control. Body and liver weights were measured before and after treatments (Table 2). Liver and body weights were significantly increased by carbon tetrachloride treatment but were decreased by DPX (40 mg/kg) and silymarin treatment. Blood was collected at the end of the experiment to assess liver enzyme levels, and AST and ALT serum levels were significantly suppressed by DPX (10 and 40 mg/kg) and silymarin treatment (Figure 4B). Gene expression of the fibrotic markers *α-SMA*, *collagen 1A1*, and *collagen 3A1* was significantly down-regulated in the carbon tetrachloride + DPX (10 and 40 mg/kg)-treated groups compared with carbon tetrachloride + PBS-treated group (Figure 4C). Similarly, the protein expression of fibronectin and α-SMA was strongly suppressed by DPX injection (Figure 4D). Moreover, H & E and Masson’s trichrome staining showed a clear reduction in collagen deposition (Figure 4E).

### 3.4. Anti-Fibrotic Effects of DPX by Inhibiting Autophagy Inhibition

As a result of a DPX mechanism study associated with HSC activation, we found that DPX inhibited autophagosome formation; HSC activation did not occur through the SMAD or MAPK pathway (data not shown). TGF-β1 increased expression of an autophagosome marker (LC3B-II) and decreased p62 expression; however, DPX treatment significantly decreased LC3B-II and increased p62 expression (Figure 5A). DPX also inhibited LC3B-II mRNA expression (Figure 5B). An autophagy flux assay was performed to determine whether reduction in LC3B-II was caused by suppression of autophagosome formation or lysosome degradation. The result of co-treatment with chloroquine, an autophagy inhibitor that prevents fusion of autophagosome and lysosome, indicated that DPX inhibited autophagosome formation. DPX decreased LC3B-II expression in cells treated with or without chloroquine (Figure 5C).

## 4. Discussion

DPM is a native plant in Korea and has been used as an anti-oxidant, anti-inflammatory, and anti-cancer agent [23]. In this study, we isolated DPX from the ethanolic extract of DPM and showed its anti-fibrotic potential in hepatic fibrosis. DPX strongly suppressed HSC activation and ECM accumulation in activated LX-2 cells and in a chronic hepatic fibrosis mouse model. Our result reveals that DPX has anti-fibrotic effects on the liver by inhibiting autophagosome formation in HSCs.

Hepatic fibrosis is characterized by excessive ECM accumulation, which is a result of chronic liver injury and wound repair processes [24]. HSCs are activated after liver injury and transdifferentiate to myofibroblast-like cells that can secrete inflammatory mediators and ECM components [5,25]. In our experiment, we used TGF-β1, one of the most well-studied cytokines, to activate HSCs [26]. TGF-β1 significantly increased expression of HSC activation markers (α-SMA and collagen) in immortalized LX-2 HSC cells. DPX reversed HSC activation in LX-2 cells at 2 µg/mL without cytotoxicity. The anti-fibrotic effect of DPX was evaluated in a chronic hepatic fibrosis mouse model induced by carbon tetrachloride intraperitoneal (I.P.) injection. In previous results, protective effects of an aqueous extract of DPM against hepatic fibrosis were reported in a thioacetamide-induced hepatic fibrosis rat model. The major components in DPM aqueous extract were reported to be syringin, chlorogenic acid, rutin, and quercetin [21]; however, DPX was found to be a major component in the dichloromethane fraction of DPM ethanolic extract. Carbon tetrachloride induces liver toxicity [27]. Cytochrome P450 enzymes metabolize carbon tetrachloride to a trichloroethane radical; these can react with nucleic acids, proteins, and lipids and result in hepatic damage and lead to inflammation, fibrosis, cirrhosis, and carcinoma associated with HSC activation [27,28,29]. In our experiment, DPX administration was started in the early stage of fibrosis after three weeks of carbon tetrachloride injections and continued for three weeks along with carbon tetrachloride administration [30]. Liver damage markers (serum AST and ALT) and fibrotic markers were significantly downregulated by DPX treatment, and its effects were comparable to those of silymarin, the hepatoprotective positive control (Figure 4).

In exploring the molecular mechanism of HSC activation, we found that DPX strongly inhibited autophagy during HSC activation. DPX induces autophagy through Erk1/2 activation in osteosarcoma cells [31]. However, in our experiment, DPX did not change the expression of Erk1/2; instead, autophagy inhibition by DPX was observed in activated HSCs (Figure 5A). Autophagy is a catabolic intracellular pathway that degrades and recycles intracellular organelles using lysosomes through the endocytic pathway [32]. Autophagic flux is increased during HSC activation, and inhibition of autophagy reverses HSC activation and hepatic fibrosis [33,34]. In this study, LC3-II expression was noticeably increased in activated HSCs, and DPX significantly suppressed LC3B-II expression and increased p62 expression (Figure 5A). p62 is a selective autophagy substrate inversely correlated with autophagic activity, because it is degraded by autophagy [35]. Light chain 3 (LC3) is a marker for autophagosome biogenesis, especially LC3-II. LC3-II is converted from LC3-I and is localized in both the outer and inner membranes of autophagosomes; in contrast, LC3-I is only localized in the cytosol. Therefore, LC3-II correlates well with autophagosome formation and is a major marker of autophagy [36,37]. However, enhanced autophagy can also decrease LC3-II through rapid autolysosomal degradation [38]. Because difficulty arises in fully explaining autophagy flux by the amount of LC3-II at a certain time point, evaluating LC3-II expression in the presence and absence of lysosomal inhibitors is important. These inhibitors block autophagosome–lysosome fusion, the last stage of autophagy. We performed an autophagy flux assay using chloroquine as a lysosome inhibitor [39]. DPX and chloroquine co-treatment inhibited autophagy by suppressing autophagosome formation.

## 5. Conclusions

To the best of our knowledge, this is the first study to show the anti-fibrotic effects of DPX in hepatic fibrosis. Hepatic fibrosis is a common chronic disease for which effective therapies are needed. Herein, we revealed that the ethanolic extract of DPM inhibits HSC activation. DPX isolated from DPM ethanolic extract reduced expression of HSC activation markers. DPX also significantly lowered serum AST and ALT levels, HSC activation markers, and collagen deposition in the carbon tetrachloride-induced chronic hepatic fibrosis mouse model. Its anti-fibrotic effect was similar to that of silymarin. Furthermore, we showed that DPX inhibited HSC activation through autophagy inhibition by measuring LC3B-II and p62 expression levels. These results support treatment potential for DPX in hepatic fibrosis by inhibiting HSC activation associated with the autophagy pathway (Figure 6).

## Figures and Tables

**Figure 1 nutrients-14-00098-f001:**
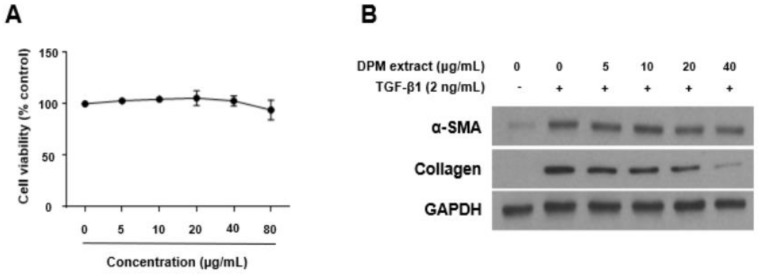
*Dendropanoxide morbifera* (DPM) ethanolic extract inhibited HSC activation and extracellular matrix (ECM) production in LX-2 cells. (**A**) Cytotoxicity was measured by the WST-1 assay after 48 h incubation with DPM extract. (**B**) α-SMA and collagen protein expression levels were analyzed by Western blot assay. LX-2 cells were activated by incubation with 2 ng/mL transforming growth factor (TGF)-β1 for 48 h and then treated with different concentrations of DPM extract for 48 h. GAPDH was used as a loading control. Each experiment was repeated three times, and values represent mean ± SD.

**Figure 2 nutrients-14-00098-f002:**
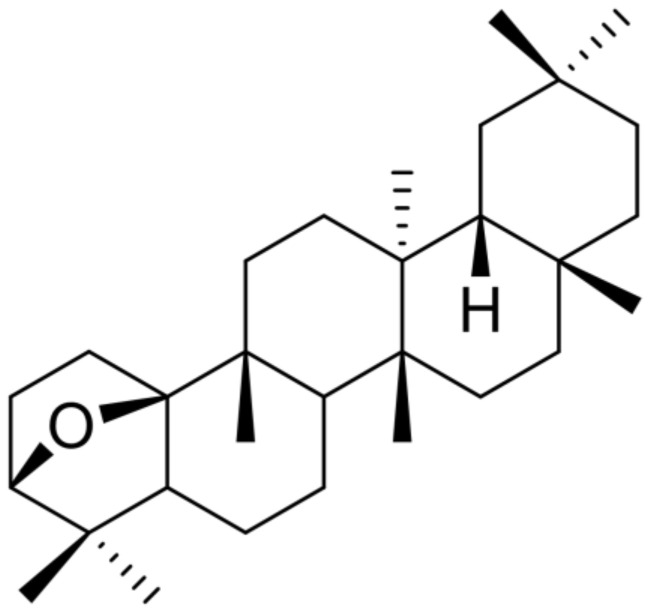
Chemical structure of dendropanoxide isolated from the ethanolic extract of DPM.

**Figure 3 nutrients-14-00098-f003:**
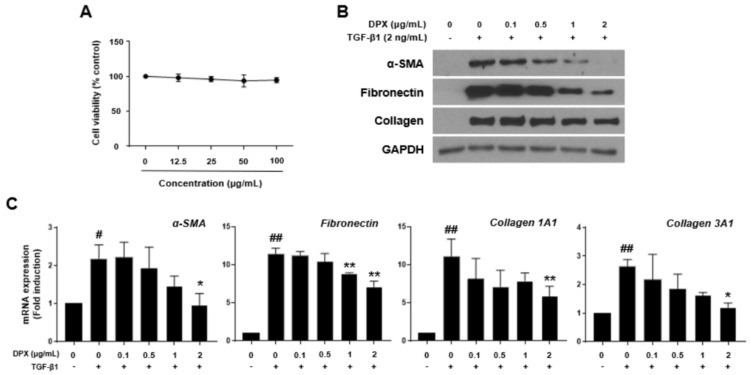
Dendropanoxide (DPX) inhibits HSC activation and collagen synthesis in LX-2 cells. (**A**) Cytotoxicity of DPX. Cytotoxicity was measured by WST-1 assay. (**B**) Protein expression of α-SMA, fibronectin, and collagen was analyzed by Western blot assay. LX-2 cells were activated by 2 ng/mL TGF-β1 for 48 h and then treated with different concentrations of DPX. GAPDH was used as a loading control. (**C**) Relative *α-SMA*, *fibronectin*, *collagen 1A1*, and *collagen 3A1* mRNA expression was analyzed by qRT-PCR analysis. Each experiment was repeated three times, and values represent mean ± SD. # *p* < 0.05, ## *p* < 0.01 compared with control, * *p* < 0.05, ** *p* < 0.01 compared with the TGF-β1 treatment group.

**Figure 4 nutrients-14-00098-f004:**
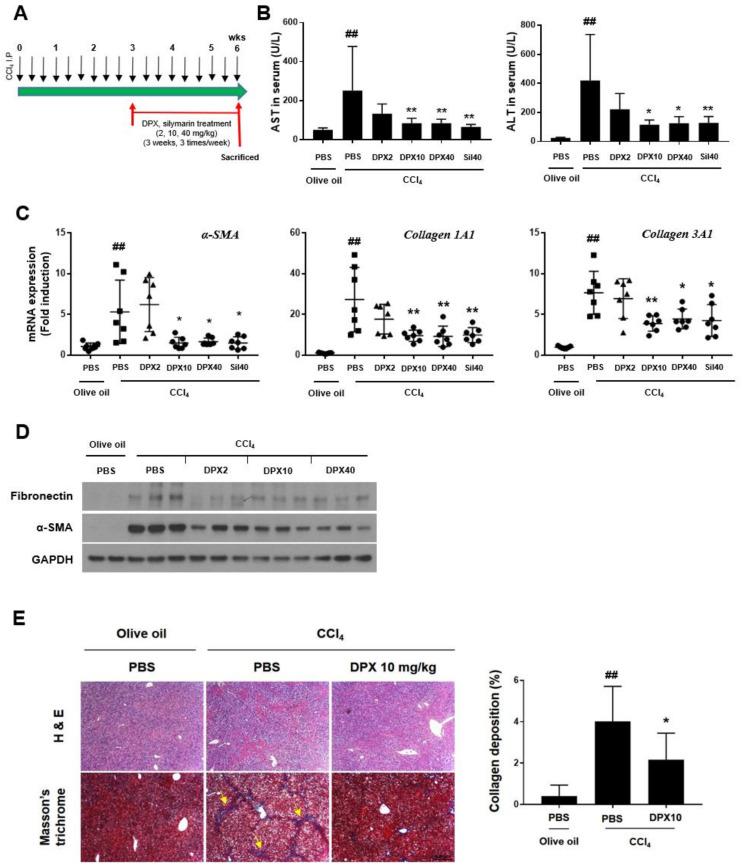
Dendropanoxide (DPX) attenuates carbon tetrachloride-induced chronic hepatic fibrosis. (**A**) Mice were treated with PBS or DPX (2, 10, 40 mg/kg) or silymarin (Sil, 40 mg/kg) for three weeks, after three weeks of olive oil or carbon tetrachloride (2 mL/kg) treatment. (**B**) Serum aspartate aminotransferase (AST) and alanine transaminase (ALT) levels were measured. (**C**) Expression of *α-SMA*, *Col1A1*, and *Col3A1* was measured by qRT-PCR analysis. (**D**) Protein expression of fibronectin and α-SMA in the livers of mice from each group was analyzed by Western blotting. GAPDH was used as a loading control. (**E**) Representative histological images of livers after H&E and Masson’s trichrome staining (yellow arrow: collagen deposition; scale bar = 500 μm). Values represent mean ± SD (*n* = 7). ## *p* < 0.01 compared with control, * *p* < 0.05, ** *p* < 0.01 compared with carbon tetrachloride treatment group.

**Figure 5 nutrients-14-00098-f005:**
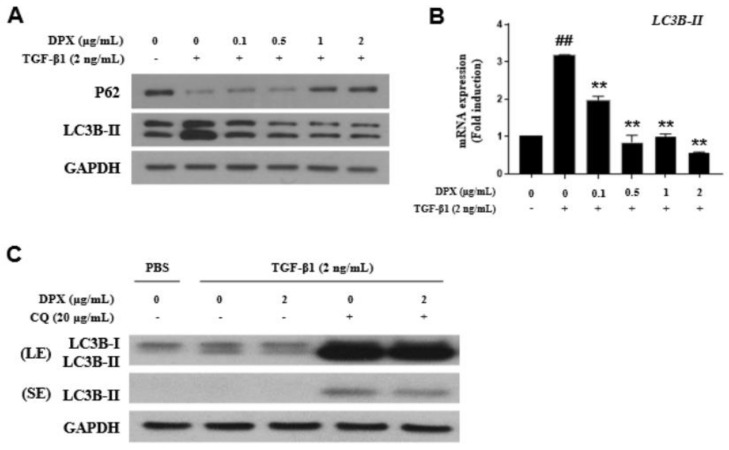
Dendropanoxide (DPX) inhibits HSC activation through autophagy inhibition. (**A**) Protein levels of autophagy markers (p62 and LC3B) were measured by Western blotting in LX-2 cells. (**B**) Relative *LC3B-II* mRNA expression was analyzed by qRT-PCR analysis. (**C**) Effects of autophagy inhibitor (chloroquine, CQ) on DPX activity were analyzed by Western blotting. GAPDH was used as a loading control. LE: long exposure, SE: short exposure. Each experiment was repeated three times, and values represent mean ± SD. ## *p* < 0.01 compared with control, ** *p* < 0.01 compared with the TGF-β1 treatment group.

**Figure 6 nutrients-14-00098-f006:**
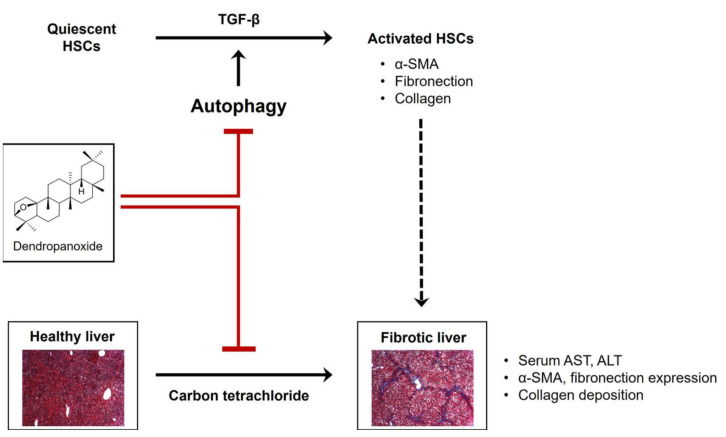
Schematic illustration of the ameliorative effect of dendropanoxide in chronic hepatic fibrosis. Dendropanoxide ameliorates chronic hepatic fibrosis by inhibiting activation of hepatic stellate cells through autophagy inhibition.

**Table 1 nutrients-14-00098-t001:** Lists of qRT-PCR primers.

Gene	Species	Forward	Reverse
*α-SMA*	Human	CTGGCATCGTGCTGGACTCT	GATCTCGGCCAGCCAGATC
*Col 1A1*	Human	GGCAACAGCCGCTTCACCTAC	GCGGGAGGACTTGGTGGTTTT
*Col 3A1*	Human	CACGGAAACACTGGTGGACAGATT	ATGCCAGCTGCACATCAAGGAC
*Fibronectin*	Human	CAGTGGGAGACCTCGAGAAG	TCCCTCGGAACATCAGAAAC
*LC3B-II*	Human	CGCACCTTCGAACAAGAG	CTCACCCTTGTATCGTTCTATTA
*α-SMA*	Mouse	GTTCAGTGGTGCCTCTGTCA	ACTGGGACGACATGGAAAAG
*Col 1A1*	Mouse	TTCGGACTAGACATTGG	GGGTTGTTCGTCTGTTTC
*Col 3A1*	Mouse	ACGTAGATGAATTGGGATGCAG	GGGTTGGGGCAGTCTAGTG

**Table 2 nutrients-14-00098-t002:** Body and liver weight in olive oil + olive oil-, olive oil + carbon tetrachloride-, DPX + carbon tetrachloride-, and silymarin + carbon tetrachloride-treated groups.

	Olive Oil + Olive Oil	Olive oil + Carbon Tetrachloride	DPX 2 + Carbon Tetrachloride	DPX 10 + Carbon Tetrachloride	DPX 40 + Carbon Tetrachloride	Silymarin 40 + Carbon Tetrachloride
Initial body weight (g)	22.78 ± 0.50	23.33 ± 1.20	23.29 ± 0.69	22.8 ± 1.06	22.0 ± 0.58	22.93 ± 1.52
Final body weight (g)	26.34 ± 0.80	25.91 ± 1.27	25.31 ± 0.69	24.36 ± 1.11	23.59 ± 0.92	24.81 ± 0.98
Liver weight (g)	1.35 ± 0.10	1.6 ± 0.17	1.51 ± 0.09	1.44 ± 0.15	1.17 ± 0.08	1.30 ± 0.12
Liver weight /body weight (×100)	5.11 ± 0.34	6.18 ± 0.49 ##	5.98 ± 0.18	5.92 ± 0.46	4.95 ± 0.32 **	5.24 ± 0.34 **

Mean + SD are shown (*n* = 7). DPX: dendropanoxide, DPX 2: DPX 2 mg/kg, DPX 10: DPX 10 mg/kg, DPX 40: DPX 40 mg/kg, Silymarin 40: silymarin 40 mg/kg. ## *p* <0.01 compared with control, ** *p* < 0.01 compared with olive oil + carbon tetrachloride-treated group.

## Data Availability

The data presented in this study are available from the corresponding author on request.

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
