# Peer review of "Dendropanoxide, a Triterpenoid from Dendropanax morbifera, Ameliorates Hepatic Fibrosis by Inhibiting Activation of Hepatic Stellate Cells through Autophagy Inhibition"

_nutrients, 2021, doi:10.3390/nu14010098_

Round 1

Reviewer 1 Report

This manuscript aims to investigate the anti-fibrotic function of dendropanoxide (DPX), isolated from Den-19 dropanax morbifera. As a potential mechanism DPX’s anti-fibrotic effects on hepatic fibrosis, the authors consider autophagy-mediated inhibition of hepatic stellate cell 20 (HSC) activation. This is an interesting study. However, the following points should be addressed appropriately before publication.

Major:

The authors used LX-2 cells to assess the role of DPX in autophagy inhibition. It is not clear whether this mechanism works in vivo. To obtain a clear understanding of autophagic activity in the fibrotic liver, it is recommended to conduct an autophagy flux assay in vivo. In vivo administration of leupeptin to mice causes accumulation of autophagosomes/autolysosomes in the liver (Kominami E, et al., J Biol Chem. 258:6093-6100; 1983.). Isolate livers from control and the fibrotic mice one hour after the injection, prepare homogenate and perform western blotting of LC3-I and LC3-II. Comparison of LC3-I and LC3-II between mice with or without leupeptin administration would provide conclusive information on in vivo autophagy flux in the liver.

The data of markers of autophagy are incomplete. It is not clear whether macroautophagy or mitophagy is the mechanism of the DPX effect. In figure 5, the material used to check autophagy is confusing. P62 is one of the mitophagy markers. For analysis of mitophagy, it is necessary to check the mitochondrial content of p62 but not whole cell lysates. This should be done. It is also very important to include western blotting data of PINK1 and parkin in addition to P62 in Figure 5C, because accumulation of PINK1 triggers Parkin recruitment to mitochondria.

Minor:

The purity of DPX used in these experiments needs to be shown.

The blood concentration of DPX in mice needs to be measured and presented.

Author Response

Response to Reviewer 1 Comments

Point 1. The authors used LX-2 cells to assess the role of DPX in autophagy inhibition. It is not clear whether this mechanism works in vivo. To obtain a clear understanding of autophagic activity in the fibrotic liver, it is recommended to conduct an autophagy flux assay in vivo. In vivo administration of leupeptin to mice causes accumulation of autophagosomes/autolysosomes in the liver (Kominami E, et al., J Biol Chem. 258:6093-6100; 1983.). Isolate livers from control and the fibrotic mice one hour after the injection, prepare homogenate and perform western blotting of LC3-I and LC3-II. Comparison of LC3-I and LC3-II between mice with or without leupeptin administration would provide conclusive information on in vivo autophagy flux in the liver.

The data of markers of autophagy are incomplete. It is not clear whether macroautophagy or mitophagy is the mechanism of the DPX effect. In figure 5, the material used to check autophagy is confusing. P62 is one of the mitophagy markers. For analysis of mitophagy, it is necessary to check the mitochondrial content of p62 but not whole cell lysates. This should be done. It is also very important to include western blotting data of PINK1 and parkin in addition to P62 in Figure 5C, because accumulation of PINK1 triggers Parkin recruitment to mitochondria.

Response: Thank you for the comments. As reviewer indicated, there are several experiments should be performed to prove the inhibitory mechanism of autophagy of dendropanoxide (DPX). However, we had spent all of DPX isolated from Dendropanax morbifera (DPM). Therefore, it is difficult to proceed further experiments now.

In this study, as a part of our ongoing projects to discover natural products showing hepatoprotective potential, we found out that DPM ethanolic extract had potential to inhibit hepatic stellate cell (HSC) activation. From the investigating of active constituents in the DPM ethanolic extract, we successfully isolated DPX which showed anti-fibrotic effects by inhibiting HSC activation. The novelty of this study is we successfully isolated DPX from DPM ethanolic extract and showed its anti-fibrotic potential in vitro and in vivo. Please consider it with this point.

Point 2. The purity of DPX used in these experiments needs to be shown.

Response: Thank you for the comment. This manuscript already had a data for the purity of DPX in the last sentence of the part “2.1. Extraction and isolation of DPX” (line 90). We used 97.6% DPX in this experiment.

Point 3. The blood concentration of DPX in mice needs to be measured and presented.

Response: Thank you for the comment. We did not check blood concentration of DPX in mice during the experiment. In other studies, using DPX in animal, we could not find the blood concentration of DPX. However, we believe that blood level of DPX in mice is enough to show hepatoprotective effects against CCl4. Because, in other studies, both DPX oral treatment (30, 60, 100mg/kg) for 14 days or DPX I.P. treatment (25mg/kg) for 5 days showed anti-diabetic effects and kidney protective effects, relatively [1,2].

  1. Moon, H.I. Antidiabetic effects of dendropanoxide from leaves of Dendropanax morbifera Leveille in normal and streptozotocin-induced diabetic rats. Hum Exp Toxicol 2011, 30, 870-875.
  2. Kim, E.S.; Lee, J.S.; Akram, M.; Kim, K.A.; Shin, Y.J.; Yu, J.H.; Bae, O.N. Protective activity of Dendropanax morbifera against cisplatin-induced acute kidney injury. Kidney Blood Press Res 2015, 40, 1-12.

Reviewer 2 Report

Fibrosis is a major concern and the authors have presented a potential candidate that can be potentially explored as a pharmaceutical agent. Very well conducted study. Minor comments that the authors may want to consider. Please see the attached file. Primarily, introduction is week and must be rewritten to build a strong case. I congratulate the authors for the completion of such a wonderful study.

Author Response

Response to Reviewer 2 Comments

Fibrosis is a major concern and the authors have presented a potential candidate that can be potentially explored as a pharmaceutical agent. Very well conducted study. Minor comments that the authors may want to consider. Please see the attached file. Primarily, introduction is week and must be rewritten to build a strong case. I congratulate the authors for the completion of such a wonderful study.

Point 1. Title; simplify the title.

Response: Thank you for the comments, however, we have a trouble to simplify the title, because we cannot use abbreviation in the title.

Point 2. Line 44; does vitamin A depletes in fibrogenic condition?

Response: Yes, in quiescent condition, vitamin A stores in HSC, however, lose stored vitamin A during HSC activation.

Point 3. Line 56, Please be specific as to what type of damage.

Response: Thank you for your comment. DPM ameliorated thioacetamide-induced hepatotoxicity and oxidative stress. We added it in revised manuscript (line 61).

Point 4. Explain more details about experiments design, relationship between HSCs, autophagy and hepatic fibrosis.

Response: Thank you for your comment. As you recommend, we added more details in revised manuscript at ‘Introduction’ (line 51-54).

Point 5. Why not test DPX 40mg/kg treated group. (Figure 4E)

Response: Thank you for your comment. In our results, anti-fibrotic effects of 10 mg/kg or 40mg/kg of DPX were similar. Therefore, we compared only CCl4 treated group and CCl4 + 10 mg/kg DPX treated group which is lower concentration treated group.

Reviewer 3 Report

In the present manuscript, Park and co-workers describe how
a triterpenoid from Dendropanax morbifera, 2 ameliorates hepatic fibrosis by inhibiting activation of hepatic stellate cells through autophagy inhibition in vitro and in vivo.

This is a comprehensive and well-developed study.
I indicate some minor comments below.

Line 25: " similar efficacy to the positive control in carbon".

Please clarify this sentence

Line 58: "activated LX2 cells". 
Indicate that these are hepatic stellate cells.

Olive oil: Please clarify if this is the vehicle used. 

Lines 111-112: "quantification. Aspartate aminotransferase (AST) and alanine transaminase 111 (ALT) levels in mouse serum were analyzed as a measure of liver function (ChemOn Inc. 112 Suwon, Korea)".
Add the procedure used

1 × 104/well 
Correct (the 4 should be in superscript). 

Specify whether the primers were designed by you or acquired from the commercial house directly. Were they from Biorad?

Line 156: What statistical analysis did you conducted with Excel?

I think section 3.2 should go before 3.1.
Figure 4E does not look too clear

Author Response

Response to Reviewer 4 Comments

In the present manuscript, Park and co-workers describe how a triterpenoid from Dendropanax morbifera, ameliorates hepatic fibrosis by inhibiting activation of hepatic stellate cells through autophagy inhibition in vitro and in vivo.

This is a comprehensive and well-developed study. I indicate some minor comments below.

Point 1. Line 25: " similar efficacy to the positive control in carbon". Please clarify this sentence

Response: Thank you for your comment. We modified it to ‘Anti-fibrotic effects of DPX were comparable to silymarine in hepatic fibrosis mouse model. We added in revised manuscript (line 27-28).

Point 2. Line 58: "activated LX2 cells". Indicate that these are hepatic stellate cells.

Response: Thank you for your comment. We added ‘human hepatic stellate’ in revised manuscript (line 65).

Point 3. Olive oil: Please clarify if this is the vehicle used. 

Response: Thank you for your comment. We added it in revised manuscript (line 112-113).

Point 4. Lines 111-112: "quantification. Aspartate aminotransferase (AST) and alanine transaminase 111 (ALT) levels in mouse serum were analyzed as a measure of liver function (ChemOn Inc. 112 Suwon, Korea)". Add the procedure used.

Response: Thank you for your comment. We added its procedure in revised manuscript (line 126-127).

Point 5. 1 × 104/well. Correct (the 4 should be in superscript).

Response: Thank you for your comment. We modified it in revised manuscript (line 131 and 149).

Point 6. Specify whether the primers were designed by you or acquired from the commercial house directly. Were they from Biorad?

Response: Thank you for your comment. All primers we used were synthesized from Bioneer (Daejeon, Korea). We added it in revised manuscript (line 144-145).

Point 7. Line 156: What statistical analysis did you conducted with Excel?

Response: Thank you for your comment. Data were analyzed using Graphpad Prism. We deleted Excel in revised manuscript (line 171).

Point 8. I think section 3.2 should go before 3.1.

Response: Thank you for your comment. Result section 3.2 is about isolation of DPX from DPM ethanolic extract. We isolated it after we evaluated the effects of DPM ethanolic extract which is result section 3.1. We believe that this order would be better.

Point 9. Figure 4E does not look too clear

Response: Thank you for your comment. We changed figure 4E in revised manuscript.

Reviewer 4 Report

The authors propose an interesting application of DPX, the work is well structured, read pleasantly and the concusions are well supported by the experiments:

- How many animals were used?
- Was DPX as well as silymarin administered in a single bolus or taken ad libitum?
- Obviously, everything should be evaluated on humans primarily for what concerns bioavailability
- How did you not evaluate a higher dosage eg 60mg?
- Could a combined use with silymarin be considered? Or other extracts known as hepatoprotective?
- You could have western originals

Author Response

Response to Reviewer 3 Comments

Point 1. How many animals were used?

Response: Thank you for your comment. We used 7 mice per group. We added it in revised manuscript (line 112).

Point 2. Was DPX as well as silymarin administered in a single bolus or taken ad libitum? Obviously, everything should be evaluated on humans primarily for what concerns bioavailability

Response: Thank you for your comment. As we mentioned at ‘Material and Methods’ (line 116-117), from day 22, mice were treated PBS or DPX or silymarin through I.P injection three times per weeks for 3 weeks.

In this study, silymarin and DPX showed similar anti-fibrotic potency at 40mg/kg. Recommendation doses of silymarin for patient with chronic liver disease is between 200mg/day to 400mg/day according to clinical trials. I assume that similar doses of DPX could be effective to chronic liver disease, however, further studies are needed before clinical trial.

Point 3. How did you not evaluate a higher dosage eg 60mg?

Response: Thank you for your comment. In preliminary study, we found out that silymarin showed significant anti-fibrotic effects at 40mg/kg I.P. injection three times per weeks for 3 weeks. Therefore, we designed 40mg/kg at the highest concentration of DPX to compare the anti-fibrotic effects of DPX with silymarin which is known to strong hepatoprotective effects.

Point 4. Could a combined use with silymarin be considered? Or other extracts known as hepatoprotective?

Response: Thank you for your comment. We are planning to evaluate the combined effects of DPX with silymarin, taurine, and caffeine which are known as hepatoprotective agents.

Point 5. You could have western originals

Response: Thank you for your comment. We submitted unmodified Western band when we submitted manuscript.

Round 2

Reviewer 1 Report

The authors did not address my comments.

Author Response

Point: The blood concentration of DPX in mice needs to be measured and presented.

Response: Thank you for the comment. According to reviewer’s comment, we checked the leftover blood samples from the animal experiment for analysis. But we found out that we used most of blood samples for previous samples. Instead, we conducted pharmacokinetic test of DPX in mice (We could use only one mouse for test. Because, as we indicated, we don’t have enough DPX for another test). The results are shown in the attached file.
